# Tubulin in Platelets: When the Shape Matters

**DOI:** 10.3390/ijms20143484

**Published:** 2019-07-16

**Authors:** Ernesto José Cuenca-Zamora, Francisca Ferrer-Marín, José Rivera, Raúl Teruel-Montoya

**Affiliations:** 1Servicio de Hematología y Oncología Médica, Centro Regional de Hemodonación, Hospital Universitario Morales Meseguer, IMIB-Arrixaca, Red CIBERER CB15/00055, 30003 Murcia, Spain; 2Grado de Medicina, Universidad Católica San Antonio (UCAM), Campus de los Jerónimos, 30107 Murcia, Spain

**Keywords:** Platelet, cytoskeleton, tubulin, platelets activation, post-translational modification, neonatal

## Abstract

Platelets are anuclear cells with a short lifespan that play an essential role in many pathophysiological processes, including haemostasis, inflammation, infection, vascular integrity, and metastasis. Billions of platelets are produced daily from megakaryocytes (platelet precursors). Despite this high production, the number of circulating platelets is stable and, under resting conditions, they maintain their typical discoid shape thanks to cytoskeleton proteins. The activation of platelets is associated with dynamic and rapid changes in the cytoskeleton. Two cytoskeletal polymer systems exist in megakaryocytes and platelets: actin filaments and microtubules, based on actin, and α- and β-tubulin heterodimers, respectively. Herein, we will focus on platelet-specific tubulins and their alterations and role of the microtubules skeleton in platelet formation (thrombopoiesis). During this process, microtubules mediate elongation of the megakaryocyte extensions (proplatelet) and granule trafficking from megakaryocytes to nascent platelets. In platelets, microtubules form a subcortical ring, the so-called marginal band, which confers the typical platelet discoid shape and is also responsible for changes in platelet morphology upon activation. Molecular alterations in the gene encoding β1 tubulin and microtubules post-translational modifications may result in quantitative or qualitative changes in tubulin, leading to altered cytoskeleton reorganization that may induce changes in the platelet number (thrombocytopenia), morphology or function. Consequently, β1-tubulin modifications may participate in pathological and physiological processes, such as development.

## 1. Introduction

Platelets, also called thrombocytes, are membrane-bound cell fragments generated and released into the bloodstream as a result of fragmentation of larger precursor cells called megakaryocytes (MKs) that originate from hematopoietic stem cells [1]. As MKs mature, they become large and polyploid. Although the mechanisms by which mature MKs release their platelets into the circulation (thrombopoiesis) is not well known, it is currently accepted that mature bone marrow MKs produce platelets via intermediate cytoplasmic extensions known as proplatelets.

Recent research has shown that platelets play a role in many diverse biological processes such us inflammation, vascular integrity, immunity, cancer metastasis and development [1]. However, their role in haemostasis and thrombosis remains one of the most important. Upon damage to the vascular endothelium, sub-endothelial matrix proteins are exposed, such as collagen and von Willebrand Factor (vWF), which provides docking sites for platelet receptors such as glycoprotein (GP) VI and GP1bα, respectively. The binding of these receptors to their ligands initiates a complex process, including several signal transduction cascades that promote platelet activation, shape change, spreading and degranulation, resulting in the recruitment of more platelets from the circulation into the platelet plug [2]. Furthermore, the activated platelet can serve as a support for interaction with coagulation factors and thus, ultimately, promote the formation of fibrin and platelet aggregate consolidation. To trigger these biological functions, platelets undergo a characteristic pattern of morphological changes in response to different stimuli. Thus, cytoskeletal reorganization during platelet activation drives the transition from a discoid platelet shape to a more spherical shape [3]. Then, platelets adopt an irregular shape thanks to the formation of pseudopods that promote adhesion and spreading on the damaged surface [3].

Cytoskeleton is not only responsible for the shape change associated with platelet activation, but also for maintaining the discoid shape of the resting platelet. Platelets have a short lifespan (7–10 days) in circulation. As a result, a continuous platelet production from MKs is required [4], and billions of platelets are produced daily. Nevertheless, under resting conditions, they maintain their typical discoid shape thanks to the presence of many well-conserved features of platelet structure: the cytoskeletal proteins. Moreover, cytoskeleton is also responsible for the shape changes associated with megakaryocyte maturation and proplatelet formation. 

Two cytoskeletal polymer systems exist in megakaryocytes and platelets: microtubules (formed by tubulins) and microfilaments (formed by actins). In this review we will focus on the first one. Following a brief review about microtubules and tubulins, we will assess the role of microtubules in mature megakaryocytes during proplatelet formation. Then, we will review the importance of microtubules in platelets under resting conditions and upon activation. Next, we will discuss how tubulin post-translational modifications may act as key regulators of microtubule cytoskeleton. Finally, we will point out how tubulin genetic variants and/or changes in tubulin expression levels may underlie a variety of platelets disorders or even physiological processes, such as development.

## 2. Platelet Cytoskeleton

Platelet cytoskeleton contains mainly spectrin, actin filaments and microtubules. Tubulins are a widely expressed protein superfamily with GTPase activity [5], comprising α, β, γ, δ, ε, ζ and η-tubulin families [6,7]. Whereas δ, ε, ζ and η -tubulin can be found only in some eukaryotic organisms; α, β and γ-tubulin are ubiquitous in eukaryotic cells. These last three tubulins are also the most abundant in human platelets, especially α and β tubulins. Particularly, the known isoforms for α-tubulin in platelets are: α1A (*TUBA1A*), α1B (*TUBA1B*), α1C (*TUBA1C*), α3C (*TUBA3C*), α4A (*TUBA4A*) and α8 (*TUBA8*). Although differences are more apparent in the carboxy-terminal tail, all these isoforms have a similar sequence and structure [8]. A copy number analysis of platelet proteins using proteomic studies showed that, compared to α and γ-tubulin, β-tubulin was more abundant both in humans and mice (Table 1) [9,10]. The human β-tubulin known isoforms in platelets are: β1 (*TUBB1*), β2A (*TUBB2A*), β2B (*TUBB2B*), β2C (*TUBB2C*), β3 (*TUBB3*), β4 (*TUBB4*), β5 (*TUBB*), β6 (*TUBB6*) and β8 (*TUBB8*) [11]. They also have more differences in the C-terminal tail [8]. Among them, β1-tubulin is the major β-tubulin isoform and is expressed exclusively in megakaryocytes and platelets [12]. Mice/human β1 tubulin is composed of 451 amino acids and its theoretical molecular weight is 50.3 kDa. As in case of other β tubulins, its N-terminal part, encompassing 113 amino acids, is involved in GTP binding and hydrolysis while middle part and C-terminal fragment are responsible for tubulin intermolecular interactions and binding of microtubule interacting proteins [13], such as MAP2 (microtubule-associated protein 2) [14,15,16]. While MAP2 is not expressed in human platelets, MAP1 is expressed and can also interact with C-terminal of β-tubulin [15]. β1-tubulin is essential for platelet microtubule formation. However, little is known about the role of specific α-tubilin isotypes. It has been reported recently, however, that an α4A-tubulin enrichment in platelets and an increase in its expression occur during the late stages of megakaryocyte differentiation [17]. 

Heterodimers of α- and β-tubulin polymerize to form microtubules. Indeed, α- and β-tubulin connect in two ways i.e., head-to-tail and side interactions. Head-to-tail interaction is needed to form protofilaments (each linear chain of heterodimers), and side interaction is required to form the microtubule, a rigid, hollow tube of approximately 25 nm in diameter. Although typically the assembly of 13 protofilaments is required to form a single microtubule, the number of protofilaments per microtubule can vary (14 or even 15) [18]. It is hypothesized that these variable numbers could be due to a rapid assembly upon activation, which it is corrected later [19]. Microtubules are polarized structures, with two functionally and structurally distinct ends (the plus end and the minus end), that differ in their affinities for tubulin dimers [20]. These polarized structures are continuously switching between phases of growth and shrinkage. In this dynamic behavior [20] seems to have a relevant role a protein family called plus-end-tracking proteins (+TIPs), a specialized microtubule-associated proteins that are conserved in all eukaryotes [21]. Although the information about the dynamics of the microtubules polymerization is limited, the expression levels from the genes that code for EB1, EB2 and CLASP1 (+TIPs proteins), in human platelets, are high (http://www.plateletomics.com/plateletomics/). In others cells, it has been described that both C-terminal and N-terminal EB1, can affect the dynamics of microtubules polymerization negatively, inhibiting the polymerizing activity [22] or by suppressing the shortening of the microtubule, respectively [23]. Additionally, CLASP1 can bind directly with EB1 and can regulate microtubule plus-end dynamics at the mammalian inner face of the cell membrane [24], acting as a microtubule-stabilizing factor. In eukaryotes, microtubules have an important role in processes such as structural support, intracellular transport, DNA segregation and cell motility. 

**Table 1 ijms-20-03484-t001:** Platelet cytoskeleton tubulin expression levels in human and mouse.

Tubulin	Molecules per Platelet (Human)	Molecules per Platelet (Mouse)
α-Tubulin	597,500 total	400,075 total
α1A	<500	-
α1B	-	54,936
α1C	174,000	25,217
α3C	110,000	-
α4A	185,000	295,013
α8	128,000	24,882
β-Tubulin	752,900 total	568,722 total
β1	144,000	246,000
β2A	-	34,850
β2B	-	284
β2C	200,000	-
β3	78,800	-
β4	96,000	227,095
β5	115,000	59,134
β6	80,000	1,359
β8	39,100	-
γ-Tubulin	2,300	796

Modified from reference [25].

Platelets are anuclear “cells” and they lack a microtubule-organizing center. The electron microscopy studies reveal that in resting platelets, microtubules are present beneath the cell membrane around the circumference of the discoid. Bundles of 3–24 microtubules form a marginal ring, also named microtubule coil [12,25], which maintains the discoid platelet shape and cell integrity in this state [11]. There is also a large amount of microtubule-binding proteins (Table 2). Some of them are involved in what is called “motor activity”. They are the microtubule-based motor proteins such as dynein, dynactin and kinesins. Dynein is regulated by dynactin and the whole dynein-dynactin complex participates in the sliding of the microtubules during platelets formation or elongation [26]. During platelet activation, this complex leads to marginal ring extension, thereby inducing the spherical shape of activated platelets [27]. Kinesins are also proteins with motor function. They participate in the transport of vesicular structures through the microtubules. Thus, kinesin-1 is involved in the mechanism of α-granules and dense granules secretion in mouse [28]. Other proteins regulate microtubule function, like CAMSAP1 (specific ‘minus-end-targeting proteins’) [29], or MAP1 (microtubule-associated protein 1 family) [30] (Table 2). In platelets, the specific function of these proteins is unclear, but they seem to play a role in microtubule assembly and stability [29,30]. On the other hand, in mouse, it has been described that members of the Ran-binding protein family, such as RanBP10, can play a crucial role both in maintaining the discoid shape in resting platelets and in platelet degranulation upon activation [31]. Finally, there are two other proteins that are important in platelets. The first one is APC (*Adenomatous polyposis coli*), which is a negative regulator of platelet biogenesis, so its blocking or deficiency promotes platelet formation [32]. The second is formin, which acts as a mediator between actin and tubulin cytoskeleton, both in platelets and megakaryocytes [33]. Formins can stabilize microtubules directly, binding to them or, indirectly, binding to end-binding proteins such as EB1 [33].”

## 3. The role of Tubulin in Platelet Formation

Platelets are produced in the bone marrow due to the maturation of megakaryocytes [1]. Derived from hematopoietic stem cells, megakaryocyte progenitors progressively lose their capacity for self-renewal and differentiate towards megakaryocytes: large and polyploid cells, present in a very low proportion in the bone marrow. Megakaryocytes undergo a process of endomitosis and become polyploid through repeated cycles of DNA replication without any division of the cytoplasm, resulting in a functional gene amplification, likely to increase protein synthesis in parallel with cell enlargement [35]. After the endomitosis is concluded, “the megakaryocyte begins a maturation process in which the cytoplasm rapidly fills itself with platelet-specific proteins, organelles, and membrane systems that will ultimately be subdivided and packaged into platelets” [36]. Over the years, several models of platelet production have been proposed, among which the formation of proplatelets has been the most accepted [36]. According to this hypothesis, bone marrow megakaryocytes produce platelets thanks to the extension of thin and long cytoplasmic protrusions (proplatelets) [37] (Figure 1) into sinusoidal blood vessels.

Prior to the onset of proplatelet formation, microtubules move from the center to the cell cortex, so that proplatelet formation begins with the development of thick pseudopods. The sliding of overlapping microtubules leads to proplatelet elongation. Microtubule assembly occurs constantly inside the whole proplatelet, including the shaft, swellings, and tip (or proplatelet ends, where there are nascent platelets) [38]. Proplatelet bending and branching amplify proplatelet ends, the entire megakaryocyte cytoplasm is converted into a mass of proplatelets, which are released from the cell, the nucleus is eventually extruded, and individual platelets are released from proplatelet ends [38]. 

Accumulating evidence throughout the years has supported the crucial role of microtubules during this process. Megakaryocytes treated with drugs that depolymerize microtubules, such as vincristine or nocodazole, do not form proplatelets [39]. A β1-tubulin knockout mouse model has shown thrombocytopenia (<50% of normal platelet count) due to a defect in the generation of proplatelets [11]. Additionally, *TUBB1* variants lead to defective platelet production, with non-functional α/β-tubulin dimers that cannot be incorporated into microtubules. [40,41,42]. In addition to *TUBB1* mutations, it has been reported that missense mutations in the α4a-tubulin coding gene cause macrothrombocytopenia, in both mice and humans [17]. Furthermore, very recently, *Tuba4a-*mutant mice have shown defects in megakaryocyte maturation and proplatelet formation [17].

Besides the microtubules’ role as the main structural components of the engine that enhances platelet lengthening, microtubules that line the axes of platelets serve a secondary function as tracks for the transport of membranes, organelles and granules into the proplatelets. Thus, individual organelles are sent from the megakaryocyte body to the proplatelets, where they move in a random and bidirectional manner, until they are captured within nascent platelets, at the ends of the proplatelets [17]. Megakaryocytes organelles seem to have a random transport [43], along a cytoskeletal track (actin filaments and microtubules), involving molecular motor proteins such as myosin, kinesin and dynein. This provides a simple and stochastic transport in order to distribute cytoplasmic contents throughout proplatelets [44]. The bidirectional transport may also be useful to spread out platelet organelles, ensuring that each platelet obtains its portion, and maybe could explain the variability in the numbers of organelles that can be found within circulating platelets [45,46].

## 4. Microtubule Organization in Resting Platelets and Upon Activation

As discussed above, the main role of the circumferential coil of the microtubule is the maintenance of the discoid shape of the resting platelet. Cross-sections of platelets show microtubules as a group of 3 to 24 circular profiles, each one of around 25 nm in diameter, at the polar ends of the lentiform cell. When discoid platelets are sectioned in the equatorial plane, the brunch of microtubules is shown under the cell wall along its greatest diameter (Figure 2) [25]. Under resting conditions, the number of tubulin dimers per platelet has been calculated to be around 250,000, 55% of which present in the polymerized form [25]. Although studies of platelet microtubule coil structure concluded that the circumferential coil is composed of a single microtubule (coiled 8 to 12 times), more recent studies suggest that, even in resting conditions, the platelet marginal band contains multiple highly dynamic microtubules of mixed polarity [47]. According to this model, the microtubule ring is formed by a single stable microtubule along with several dynamic microtubules that polymerize in both directions around the marginal band. In this way, microtubules continuously assemble around the periphery of the resting platelet [47]. The importance of the microtubule coil in resting platelet shape and size has been shown through different experiments, i.e., cooling [1], disrupting agents [1] and β1-tubulin knockout mice [11], where platelets become more spherical due to microtubule depolymerization. In fact, β1-tubulin mutations result in spherical platelets [40,41,42]. Moreover, people carrying the p.Q43P variant in β1-tubulin in heterozygosis have an elevated portion of platelets with a spherical shape, due to lower β1-tubulin expression levels [48]. Altogether, these studies strongly support the role of the marginal coil in maintaining platelet discoid appearance.

Following activation, changes in the location and organization of the microtubule occur in order to allow the transition from the discoid to a spherical shape. Thus, during platelet activation, microtubule coils become compressed into the center of the platelets prior to their fragmentation and depolymerization (Figure 2) [47]. This happens because in resting platelets, the microtubule coil is under the influence of kinesin and dynein motor proteins, which act antagonistically and result in no net movement of the microtubules between each other [25]. Upon activation, however, the high intracellular calcium concentration enhances dynein activity and reduces kinesin activity, allowing dynein walking and microtubule movement past each other, causing the microtubule coil to expand (Figure 2) [25]. Because of the limited space within the platelet, the expanding microtubules fold and coil into the center of the activated platelet, changing the platelet morphology towards a spherical shape (Figure 2) [25,27]. Following the expansion and centralization of the microtubules, the microtubule coil finally will depolymerize and form the network of microtubules observed in spread platelets (Figure 2). The exact function of the microtubules in spread platelets is unknown [25]. 

## 5. Tubulin Post-Translational Modifications in Platelets

Post-translational modifications are dynamic and often reversible processes where the addition of a chemical group to the amino acid residues of a protein induces changes in the protein’s functional properties. Tubulin, as with most proteins, can undergo post-translational modifications, which can generate microtubule diversity. The most relevant post-translational modifications that can appear in tubulin are acetylation, phosphorylation, detyrosination/tyrosination, polyglutamylation and polyglycylation [8]. These post-translational modifications are one of the factors that affect the properties of the tubulin polymer and allow the microtubules to adapt to the different functions that they have in the cells. 

In platelets, the ability to rearrange the microtubule coil seems to be related to the post-translational modifications of α-tubulin, including acetylation (in Lys40 or K40) and detyrosination/tyrosination processes (in C-terminal tyrosines). “Another existing post-translational modification of α-tubulin detected in platelet marginal band is Δ2-tubulin [49], which consists of removing sequentially the two C-terminal amino acids, i.e., tyrosine and glutamate [8]. Thus, the previously tubulin detyrosination modification is required [50] and once Δ2-tubulin happens, C-terminal cannot be retyrosinated [50].” While stable microtubules are characterized by acetylation [47,51] and detyrosination, more dynamic microtubules are tyrosinated and deacetylated [52]. Importantly, most platelets contain both dynamic and stabilized microtubules, but the abundance of either isoform appears to vary within a given platelet [47]. 

At a functional level, the acetylation status of the microtubule may regulate microtubule motor activities [25,53]. The enzymes involved in K40 deacetylation are HDAC6 (histone deacetylase family member 6) and SIRT2 (sirtuin 2) [53,54]. Following platelet activation, microtubules are subject to a HDAC6-mediated deacetylation. Although deacetylation is not crucial for platelet spreading, the capacity of HDAC6 to prevent tubulin hyperacetylation influences the speed of platelet spreading [55]. Furthermore, the post-translational modification of the microtubules may help to modulate microtubule dynamics upon platelet activation. Thus, upon platelet activation, the marginal band becomes heavily polyglutamylated, which allows the mobilization of motor proteins, such as dynein, to achieve the shape change required during platelet activation [25]. Moreover, in mature megakaryocytes and proplatelets, the polyglutamylation of β1 tubulin-containing microtubules is important in order to induce the formation of the marginal band [56], and it is implicated in proplatelet elongation and in platelet release [57]. A list of the proteins that regulate microtubule organization and dynamics is shown in Table 2.

## 6. Implications of Tubulin Genetic Variability in Pathological and Physiological Platelet Function

In recent years, several mutations in β1-tubulin (a minor allele frequency [MAF], ≤ 1%) have been identified, such as c.952C>T (p.Arg318Trp); c.779T>C (p.Phe260Ser) and c.1267C>T (p.Gln423*) [13]. All these mutations are associated with macrothrombocytopenia due to impaired proplatelet formation. Although microtubules are decreased, disorganized and did not form platelet marginal band, there is no abnormal bleeding [13]. 

The first functional *TUBB1* genetic variant reported in humans was the double nucleotide substitution c.128_129delAGinsCC, predicting p.Gln43Pro in the β1-tubulin chain. With a MAF of 7.7%, this is a common variant identified in 24% of 33 patients with unexplained macrothrombocytopenia, but also in 11% of controls. This polymorphism was associated with enlarged round platelets with defective marginal bands. Based on an association study, we previously reported that β1-tubulin p.Q43P polymorphism is a weak risk factor for intracranial bleeding [58], but does not protect against acute coronary syndrome [59]. 

At least two other common genetic variants (MAF > 1%) have been reported: *TUBB1* c.821C>T (p.Thr274Met) and *TUBB1* c.920G>A (p.Arg307His). The first one has been associated with less thrombocytopenia under paclitaxel treatment. The second one has been associated, compared to wild type *TUBB1*, with a lower platelet count in Bernard-Soulier Syndrome and in patients with immune thrombocytopenia [13]. Overall, although common population *TUBB1* variants are associated with a weak phenotype effect, they may worsen the phenotype of other inherited or acquired platelet disorders that modify platelet number or size [13]. 

However, tubulin does not only have a role under pathological conditions. Age-related physiological changes are not only present in secondary haemostasis, that is, in the plasma levels of coagulation factors [60], but also in platelets. Compared to adult platelets, platelets from neonates are hyporeactive to most physiological agonists and show an impaired agonist-induced secretion [61]. Upon activation, the contraction of the platelet marginal band induces the centralization of granules (Figure 3A) [62]. Despite the absence of differences between neonatal and adult platelets under resting conditions, our group has recently shown that, upon stimulation, adult platelets exhibited dramatic alterations in shape and noticeable degranulation. Neonatal platelets, in contrast, displayed minor shape changes and their granules remained clearly visible throughout the cytoplasm, suggesting a limitation in the granule centering process (Figure 3B) [63]. 

Since the centralization of granules is a process that depends on the contraction of the microtubule marginal coil [31,64], and β1-tubulin is its main isoform (up to 90% of total β tubulin in megakaryocytes and platelets), we investigated whether β1-tubulin expression is developmentally regulated. Our findings showed a significant reduction in β1-tubulin mRNA (*TUBB1*) (~40%) and protein expression levels in neonatal compared to adult platelets (Figure 4A,B with permission) [63]. Neonatal platelets, however, displayed a microtubular ring similar to the present one in adult platelets (Figure 4C,D with permission) [63], suggesting that half the usual amount of β1-tubulin may be sufficient to allow the characteristic discoid shape. Our results are in agreement with those reported for p.Q43P β1-tubulin variant homozygous carriers, whose platelets, despite having half the normal levels of β1-tubulin, mostly displayed a normal marginal band and a discoid shape [63,65]. These homozygous PP carriers showed, however, a small percentage of platelets without a clear microtubular ring, but rather they showed a diffuse β1-tubulin in the cytosol [65]. The fact that we do not observe the disperse β1-tubulin in the cytosol from neonatal platelets may be due to the overexpression, in neonatal platelets, of the other β-tubulin isoforms: *TUBB2A* and *TUBB* (Figure 4E,F with permission) [63]. Overall, our findings showed that β1-tubulin is developmentally regulated. Although these developmental differences can be overcome by physiologic compensatory mechanisms in healthy full-term neonates, these mechanisms may work inefficiently under stress conditions (i.e., prematurity, sepsis) and contribute to an increased risk of bleeding and thrombocytopenia among sick neonates [63].

In summary, it is well established that microtubules, formed by α/β tubulin heterodimers, are key actors in the formation of proplatelets from mature megakaryocytes, preceding nascent platelets. In platelets, microtubules form a subcortical ring, the so-called marginal band, that preserves the elliptic shape of resting platelets and play a critical role in cytoskeleton re-organization during platelet activation. Among the large amount of microtubule-binding proteins, dynein, dynactin and kinesins contribute to the so called “motor activity”, that drives sliding of the microtubules during platelets formation and activation. Moreover, recent data indicate that microtubule function is regulated by many others microtubule interacting proteins such as MAPs, EB, Clasp, RanBP, APC and formins. Post-translational modifications of tubulin (acetylation, phosphorylation, detyrosination/tyrosination, polyglutamylation and polyglycylation) account for a great microtubule diversity allowing the microtubules to adapt to the different cell functions. In addition, tubulin is genetically diverse and expression of variant forms of tubulin, from relatively common to very rare variants, gives rise to variability in platelet size in healthy individuals and rare inherited thrombocytopenia. Finally, we have shown that tubulins are developmentally regulated, and this seems to be a natural mechanism contributing to the well established hypo reactivity of neonatal platelets.

## Figures and Tables

**Figure 1 ijms-20-03484-f001:**
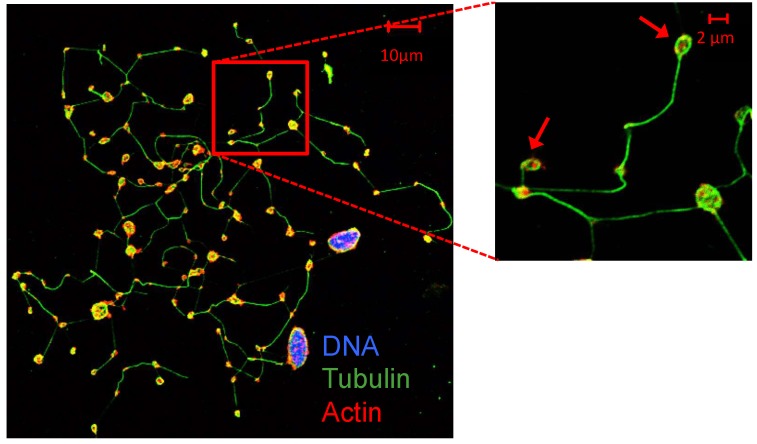
Megakaryocytes and proplatelets [37]. Human primary megakaryocytes and proplatelets derived from CD34^+^ hematopoietic stem cells progenitors obtained from cord blood (CB). At day 14 of culture, megakaryocytes show proplatelet (pointed with arrows) formation. DNA (DAPI, purple-blue), α-tubulin (green) and actin (red) staining.

**Figure 2 ijms-20-03484-f002:**
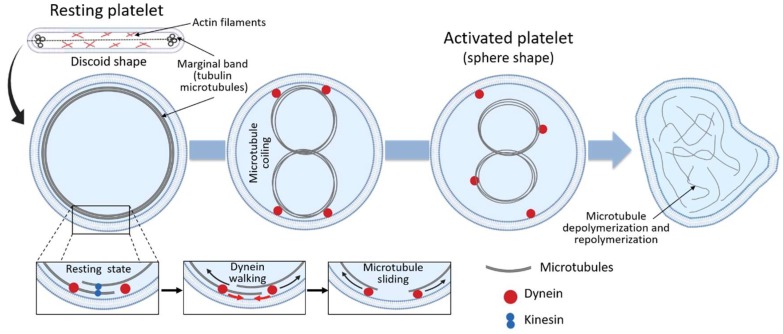
Microtubules coiling during platelet activation.

**Figure 3 ijms-20-03484-f003:**
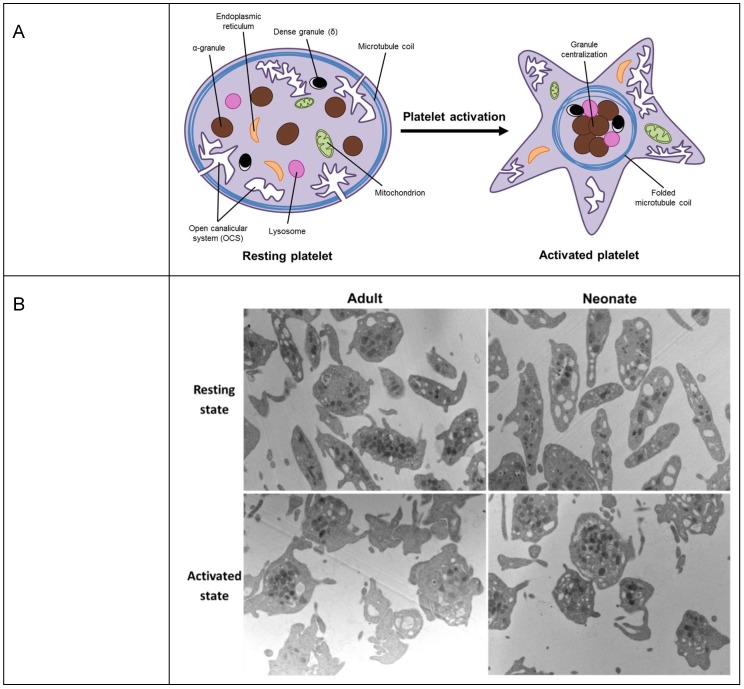
Platelets under resting and stimulation conditions. (**A**) A schematic representation of granule reorganization during platelet activation. (**B**) Neonatal and adult representative platelet ultrastructure images, obtained by electron microscope with a magnification of 9700×. Granule centralization, before and after stimulation, was quantified in 50 sections per sample (3 samples per group), and per state (basal and stimulated). Statistical analysis showed differences in granule distance to the centroid in adult basal platelets as compared to 10 µM Thrombin receptor agonist peptide-6 (TRAP)-stimulated adult platelets (*p* < 0.01). No differences were found, by contrast, between basal and stimulated neonatal platelets or between quiescent adult and neonatal platelets. Granule centralization was measured using the Leica QWin Pro v3 Software (Leica Microsystems).

**Figure 4 ijms-20-03484-f004:**
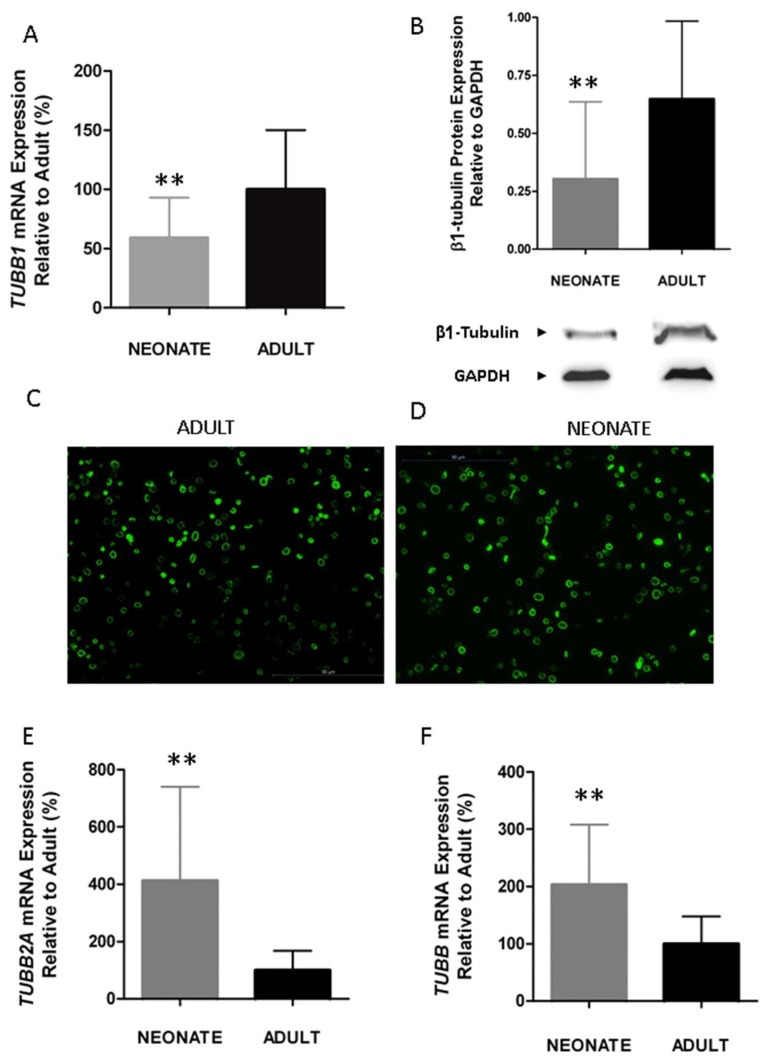
Neonatal platelets display a reduced expression of β1-tubulin but an overexpression of other isoforms (*TUBB2A* and *TUBB*), allowing a normal marginal band. (**A**) *TUBB1* mRNA level measured by quantitative real time qRT-PCR and normalized with respect to *ACTB* (*n* = 21/group). (**B**) Quantification of β1-tubulin protein levels by immunoblotting. Western blot pictures shown are representative images cropped from two gels performed at the same time and under the same conditions (*n* = 15 samples/group). Quantification was performed by densitometric analysis using the ImageJ program, and was normalized to β-Actin. (**C**,**D**) Microscopy of adult and neonatal platelet marginal bands using α-tubulin immunofluorescence in adhered platelets on poly-l-lysine. (**E**,**F**) Quantification of *TUBB2A* and *TUBB* mRNA levels by qRT-PCR (*n* ≥ 10 samples per group). ** *p* < 0.01. Taken from [63] with permission.

**Table 2 ijms-20-03484-t002:** Major platelet microtubule- interacting proteins.

Protein	Principal Function in Platelets
Tubulin α, β, γ	The microtubule coil helps to maintain the resting platelet discoid shape. Upon activation, the microtubule coil centralizes, allowing shape change, granule release and spreading.
Dynein	“A subunit of the microtubule motor complex that drives the sliding of the microtubule coil during platelet activation”.
Dynactin	A subunit of the microtubule motor complex that increases the processivity of the dynein motor.
Kinesin 1 and 4	“The movement of granules/organelles along microtubules. Antagonizes dynein motor action in resting platelets”.
Microtubule-associated proteins (MAPs) 1 and 4	“Involved in microtubule assembly. Its roles in platelets are unknown”.
End-binding protein (EB)	“Binds to the microtubule plus ends to regulate microtubule activity, through post-translational modifications in EB protein itself [34].”.
CLIP-associating protein (Clasp)	“Possibly plays a role in microtubule organization”.
Ran-binding protein (RanBP)	“The regulation of microtubule coil organization and contraction”.
Stathmin	“Microtubule disassembly. No confirmed role in platelet activation”.

Modified from reference [25].

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
