# Peer review of "Tubulin in Platelets: When the Shape Matters"

_ijms, 2019, doi:10.3390/ijms20143484_

Round 1
Reviewer 1 Report
The submitted review (Tubulin in platelets: when the shape matters) requires major reorganization and corrections. I would recommend first to introduce megakaryocytes and pro-platelets and platelets (lines 99-111), next provide information about the biological processes in which platelets are involved, then present information about the role of a cytoskeleton in these processes, point that Authors would like to focus on microtubules and present a general information about tubulin and microtubules. Next, I would suggest focussing on microtubules in megakaryocytes and their changes/ rearrangement specifically during proplatelets formation and in platelets (resting and activated).
Chapters 2 and 6 are well written while an introduction and chapter 1 are poorly written, with numerous mistakes and require editing by a native speaker.
Some information is missing. For example, for a Reader who is not familiar with the platelets it would be interesting to learn that platelets do not have microtubule organizing center, more detailed information could be provided about the role of a4-tubulin and b1-tubulin, tubulin glutamylation was not mentioned (van Dijk et al., 2018), role of APC protein (Strassel et al., 2018), role of formins (Zuidscherwoude et al., 2019).
In the end, more general conclusion concerning the role of microtubules in platelets would be useful.
I regret but due to issues mentioned above and suggested corrections (details below) as it is now I can’t recommend this manuscript for publication. However, because the topic is interesting I would encourage Authors to correct and re-submit their work.
Abstract:
Platelets are anuclear cells that play an essential role in many pathophysiological processes including haemostasis, inflammation, infection, vascular integrity, and metastasis. Platelet lifespan is short and, billions of platelets are produced daily from megakaryocytes.
Suggested change:
Platelets are anuclear cells of a short lifespan that play an essential role in many pathophysiological processes including haemostasis, inflammation, infection, vascular integrity, and metastasis. Billions of platelets are produced daily from megakaryocytes.
When platelets became activated, however, dynamic and rapid changes occur in cytoskeleton structure.
Suggested change:
Activation of platelets is associated with dynamic and rapid changes in the cytoskeleton.
Two cytoskeletal polymer systems exist in megakaryocytes and platelets: actin filaments and microtubules, based on actin and tubulin monomers, respectively.
Comment:
Microtubules are composed of a-, b-tubulin heterodimers not tubulin monomers.
Line 16: Focus in - should be “focus on”
Line 16: 1-tubulin – beta symbol disappeared, similar a-tubulin (next line)
During thrombopoiesis, microtubules mediate proplatelet elongation and granule trafficking from megakaryocytes
Comments: Please, first introduce a term “megakaryocytes”
Molecular changes, either polymorphism or rare variants….
Comments: What do you mean by this statement?
…leading to cytoskeleton reorganization and altered platelet formation, morphology or function.
Comments: What do you mean by “altered platelet formation”?
… modifications may be involved in both pathological…
Suggested change:
…modifications may cause both pathological
Introduction:
Line 30: Platelets (thrombocytes) are…
Line 34: …several signal transduction cascades, morphological transformations that promote platelet activation, cytoskeletal reorganization, spreading…
Comment: causes and consequences are mixed here. Can morphological transformation promote platelet activation?
Line 41: is: pathophysiological, should be pathophysiological
Line 43: is: please correct this sentence
Line 47-48: please correct this sentence
Line 49-50: an importance of “microtubules” not “tubulin”. Tubulin as a monomer can not maintain a shape of the cell.
Line 52: what Authors understand by “tubulin variants”?
Line 53: “ may be involved in: does not sound right here
1. Platelets cytoskeleton: tubulin
Line 55: Cytoskeleton contains spectrin, actin filaments and microtubules, actin and tubulin are building blocks.
Line 56-57: What Authors understand by “ shared or with more specific function”. This information is not clear.
Table 1.
Title: rather: “microtubule interacting proteins”
a-, b-, g- tubulins are not tubulin isoforms
dynein and dynactin are subunits of the complex
listed kinesins, MAPs, are not examples of isoforms.
Line 59: tubulin superfamily not family
Line 60: first, microtubules are implicated in the mentioned processes, not tubulin, second Authors mentioned “exocytosis”, but they probably meant “ intracellular transport”
Line 61-62: a,-b-g-tubulin are....in eukaryotic cells; ….tubulin can be found in other organisms, such as protists.
First, protists are Eukaryotes, second, for example d-tubulin is in centrosomes in mammals. So this sentence is incorrect and contradicts a next sentence (line 63).
Line 67: most abundant tubulin, not tubulin isoforms. Please see also below, and correct usage of “isoform” and “isotype”.
Line 69-70: in case of tubulin the C-terminal end is called “tail” not “coil” (also line 75).
Line 71: Copy number analysis of platelet proteins, by using …(remove “by”)
Line 75: cited REF 8 describes g-tubulin, not differences of the amino acid sequence of the tail of b-tubulin isotypes. Please cite proper REF. Also, line 70 (a-tubulin).
Line 78: please see Strassel et al., 2019, Life Science Alliance (An essential role for α4A-tubulin in platelet biogenesis)
Line 79: Microtubules are polymers built from dimers of a- and b-tubulin (protofibrils) that…
First, microtubule is composed of protofilaments not protofibrils. Second, it suggests that dimers are protofibrils. To form a tube there are two types of interactions between heterodimers (head-to-tail and side interactions).
Line 81: 13 protofilaments are most common, but MTs with a different number of protofilaments also exist.
Line 83: dimer not dimmer
Line 83-84: “The way that these subunits shape those polymers and their architecture, is different across species” – what Authors mean by this statement?
Line 84-86 – repetition. Importance of microtubules (as importance of “tubulin”) in diverse biological processes was mentioned earlier.
Table 2: it is stated that it is a modification of work from REF 14 (Schwer et al., 2001). I can not find a Table in this REF.
Lines: 90-94:
Authors should be aware, that adding words” whereas” and “and” as well as changing where the sentence ends in somebodies text, is not proper scientific writing. In such a case, the text should be cited literally and taken in quotation marks. Otherwise, it could be considered plagiarism.
For comparison text from Burley et al., 2018
“Tubulinβ-1 chain is a 50.3 kDa protein containing 451 amino acids. The amino-terminal region (residues 1–206) mediates GTP binding and hydrolysis which is necessary for microtubule assembly. The intermediate region (residues 207–384) interacts with other tubulin molecules within the same or adjacent protofibrils. The carboxyl-terminal region (residues 385–45) mediates microtubule assembly and interactions with regulators [2,3].”
Lines 95-97 – why these two proteins were mentioned (CAMSAP1 and MAP1)? Are they important in platelets? Publication cited by Authors refer to a general role of these proteins.
2. Role of tubulin in platelets formation
Line 109: “according to this hypothesis” instead of “In this theory”
Line 141: tubulin microtubules – change to microtubules (all microtubules are from tubulin)
Line 141: involving also…. Is “also” necessary?
Lines 143-146: The bidirectional transport may also be useful to mix platelet organelles, ensuring that each platelet obtains its portion, and maybe could explain the variability in the numbers of organelles that can be found within circulating platelets [31,32].
I am not sure about the usage of the word “mix” here.
3. Microtubules organization in resting platelets
Line 150:
Is: ….studies reveal that resting platelet microtubules are found around the circumference……form a marginal ring
Change to: ….studies reveal that in resting platelet microtubules are present beneath cell membrane around the circumference….where they form a marginal ring…
Line 153: microtubules “sit” (?), clear link – suggestion: an apparent link
Line 155-156- repetition, please remove.
Lines 156 and next – refers to chapter 1.
Suggestion: please combine chapter 1 and chapter 3. Please clearly state that microtubules ring is also named microtubules coil (explain why coil – microtubules arrangement). In case of C-terminal end of tubulin use a term “tail”.
5. Tubulin post-translational modifications in platelets
Line 189: remove: as we have just seen” What Authors mean by “MAPS can modulate this adaptation” – please be more precise.
General remark: please clearly separate general information regarding tubulin posttranslational modification and data concerning microtubules in platelets, cite appropriate publications. Publication about microtubules acetylation in resting cells is not cited here (REF 35, Patel-Hett et al., 2008).
Line 201-202: Enzymes involved in K40 deacetylation are HDAC6 (histone deacetylase family member 6).
Enzymes (two), HDAC6 and SIRT.
van Dijk et al., 2018, “Microtubule polyglutamylation and acetylation drive microtubule dynamics critical for platelet formation” It should be cited and role of glutamylation mentioned.
6. Implications of tubulin genetic variability in pathological and physiological platelet function
Please explain what do you mean by rare variants, how it refers to tubulin mutation.
Lines 212-213:
…and loss of microtubule activity (absence of the platelet marginal band), but…
Loss of activity means that MTs are present but are not active (how Authors understand MTs activity, motor proteins activity on microtubules?). Absence of band / ring means that there are no MTs – if something is not present it can not be active. Please re-write.
Lines 233-238: Despite the absence of differences between neonatal and adult platelets under resting conditions, our group has recently shown that upon stimulation, MISSING TEXT ? while..
Sentence is very long. Please divide this sentence.
Lines 249-250: Since centralization of granules is a process that depends on contraction of the microtubule marginal coil,…..
Contraction or transport via motors, please explain (please cite REF)
Author Response
Reviewer #1 Comments and Suggestions for Authors
The submitted review (Tubulin in platelets: when the shape matters) requires major reorganization and corrections. I would recommend first to introduce megakaryocytes and pro-platelets and platelets (lines 99-111), next provide information about the biological processes in which platelets are involved, then present information about the role of a cytoskeleton in these processes, point that Authors would like to focus on microtubules and present a general information about tubulin and microtubules. Next, I would suggest focussing on microtubules in megakaryocytes and their changes/ rearrangement specifically during proplatelets formation and in platelets (resting and activated).
Authors’ Response: We thank the reviewer his comment to improve the review organization. Thanks to this comment we realized that the review had to be reorganized in another way in order to make it clearer. As suggested by the reviewer, first we introduced megakaryocytes and pro-platelets (first paragraph of the introduction in the new version of the manuscript); then we focus on platelets and the biological processes in which are involved (second and third paragraphs of the new introduction); and finally, the role of a cytoskeleton and microtubules in platelets (fourth and fifth paragraphs of the introduction in this new version).
To address comments from reviewers 1and 2; two major changes have been made.
A. We have moved the “microtubule coils” concept to the first chapter of the manuscript (platelet cytoskeleton); just after we provide information about the structure and function of the microtubules and following the Table 1. In this way, the “microtubule coils” concept is explained before it was mentioned.
B. We have reorganized the previous sections “Microtubules organization in resting platelets” and “Microtubules reorganization in platelets activation” into a new merged section “Microtubules organization in resting platelets and upon activation” (chapter 3 of the new version of the review):
Altogether, chapter 1 and Introduction have been modified deeply.
Chapters 2 and 6 are well written while an introduction and chapter 1 are poorly written, with numerous mistakes and require editing by a native speaker.
Authors’ Response: We thank the reviewer his comment that raise the issue concerning to the English and grammar. Knowing the limitations of our English writing skills, the paper was reviewed by the International Journal of Molecular Sciences English Editing Services.
Some information is missing. For example, for a Reader who is not familiar with the platelets it would be interesting to learn that platelets do not have microtubule organizing center, more detailed information could be provided about the role of a4-tubulin and b1-tubulin, tubulin glutamylation was not mentioned (van Dijk et al., 2018), role of APC protein (Strassel et al., 2018), role of formins (Zuidscherwoude et al., 2019).
Authors’ Response: We appreciate this suggestion from the reviewer, and have incorporated this information into the revised version of the manuscript. As suggested by the reviewer, a mention that platelets do not have microtubule organizing center was added (page 3, first line after Table 1). Information about the role of a4-tubulin and b1-tubulin (Page 2, last paragraph, just between # ref 13-14), tubulin glutamylation (chapter 4, last paragraph, ref # 43) , APC protein and formins (pag 3, last paragraph and reference # 21 and 22), along with the new references have been incorporated into the review.
In the end, more general conclusion concerning the role of microtubules in platelets would be useful.
Authors’ Response: As suggested by the reviewer, more general conclusion concerning the role of microtubules in platelets was added at the end of the review.
I regret but due to issues mentioned above and suggested corrections (details below) as it is now I can’t recommend this manuscript for publication. However, because the topic is interesting I would encourage Authors to correct and re-submit their work.
Authors’ Response: We fully appreciate that the reviewer considers the topic interesting and gives us the opportunity to resubmit it once we have addressed all the issues mentioned above as well as the suggested corrections listed below.
Abstract:
Platelets are anuclear cells that play an essential role in many pathophysiological processes including haemostasis, inflammation, infection, vascular integrity, and metastasis. Platelet lifespan is short and, billions of platelets are produced daily from megakaryocytes.
Suggested change:
Platelets are anuclear cells of a short lifespan that play an essential role in many pathophysiological processes including haemostasis, inflammation, infection, vascular integrity, and metastasis. Billions of platelets are produced daily from megakaryocytes.
Authors’ Response: We appreciate this suggestion from the reviewer, and have incorporated it in the new version of the manuscript.
When platelets became activated, however, dynamic and rapid changes occur in cytoskeleton structure.
Suggested change:
Activation of platelets is associated with dynamic and rapid changes in the cytoskeleton.
Authors’ Response: The suggested correction was taken in account and we have translated the comment to International Journal of Molecular Sciences English Editing Services in order that the new version is reviewed carefully.
Two cytoskeletal polymer systems exist in megakaryocytes and platelets: actin filaments and microtubules, based on actin and tubulin monomers, respectively.
Comment: Microtubules are composed of a-, b-tubulin heterodimers not tubulin monomers.
Authors’ Response: The suggested correction was taken into account and the text was modified accordingly.
Line 16: Focus in - should be “focus on”
Authors’ Response: The suggested correction was taken into account and the text was modified accordingly.
Line 16: 1-tubulin – beta symbol disappeared, similar a-tubulin (next line)
Authors’ Response: The suggested correction was taken into account and the text was modified accordingly.
During thrombopoiesis, microtubules mediate proplatelet elongation and granule trafficking from megakaryocytes
Comments: Please, first introduce a term “megakaryocytes”
Authors’ Response: We appreciated the reviewer comment and megakaryocytes term was introduced in “Abstract” section. Thus, the following text has been added in “Abstract” section (Pag. 1):“… Billions of platelets are produced daily from megakaryocytes (platelet precursors).
Molecular changes, either polymorphism or rare variants….
Comments: What do you mean by this statement?
Authors’ Response: We appreciated the reviewer comment in order to clarify the Abstract. The sentence was not accurate. We have modified the manuscript, and the following text has been added in “Abstract” section (Pag. 1): “Molecular alterations in the encoding gene” ……
…leading to cytoskeleton reorganization and altered platelet formation, morphology or function.
Comments: What do you mean by “altered platelet formation”?
Authors’ Response: We agree appreciated the reviewer comment. The sentence was not accurate. We have modified the manuscript and the following text has been added in “Abstract” section (Pag. 1): “…leading to altered cytoskeleton reorganization that may induce changes in the platelet number (thrombocytopenia), morphology or function.”
… modifications may be involved in both pathological…
Suggested change: …modifications may cause both pathological
Authors’ Response: The suggested correction was taken into account in the new version.
Introduction:
Line 30: Platelets (thrombocytes) are…
Authors’ Response: The suggested correction was taken into account and the text was modified accordingly. The statement can be found in: pag 1, “introduction” chapter, line 1.
Line 34: …several signal transduction cascades, morphological transformations that promote platelet activation, cytoskeletal reorganization, spreading…
Comment: causes and consequences are mixed here. Can morphological transformation promote platelet activation?
Authors’ Response: We appreciated the reviewer comment. We clarified, in this revised version, that the platelets, through their membrane receptors, respond to different external ligands and this triggers a series of changes in the platelet, starting with its activation, following with its change of form, release of the granular content and ending with the thrombus formation. We did not want to say that morphological transformation promotes platelet activation. The previous sentence was not fully accurate. We have re-written this paragraph in order to be clearer. This clarification can be found in the second paragraph of the “introduction” chapter (at the end of page 1 and at the beginning of page 2)
Line 41: is: pathophysiological, should be pathophysiological
Authors’ Response: The suggested correction was taken into account and the text was modified accordingly (pag 2, second line)
Line 43: is: please correct this sentence. Original sentence: “Because platelets lifespan is no more than 10 days in circulation ”,
Authors’ Response: The suggested correction was taken into account and the text was modified accordingly. Thus, the following text has been added:” Platelets have a short lifespan (7–10 days) in circulation. As a result,…..” The new statement can be found in the second paragraph of page 2 (or third paragraph of the introduction).
Line 47-48: please correct this sentence. Original sentence: “thanks to the presence of many well-conserved features of platelet structure. Platelet morphology is important in the platelet biological key process: from its own formation, going through its resting state to its activation.”
Authors’ Response: The suggested correction was taken into account and the text was modified accordingly. The new statement can be found at the end of the second paragraph of page 2.
Line 49-50: an importance of “microtubules” not “tubulin”. Tubulin as a monomer can not maintain a shape of the cell. Original sentence: In this review we will explain the importance of one of the cytoskeleton´s main constituents: the tubulin.
Authors’ Response: We fully agree with the reviewer, and have clarified this in the manuscript. The revised statement can be found in page 2, beginning of the third paragraph.
Line 52: what Authors understand by “tubulin variants”?
Authors’ Response: It simply refers to a change in the gene sequence. There is recommendation for naming all changes in gen sequence as Variants and then sub classifying them as one of pathogenic, likely pathogenic, uncertain significance, likely benign, or benign (Genet Med. 2015 May; 17(5): 405–424. doi:10.1038/gim.2015.30).
Line 53: “ may be involved in: does not sound right here
Authors’ Response: In response two last three comments, and following the suggestion of Reviewer 2, we have modified the entire last introduction paragraph. This specific statement can be found on page 2 of the revised manuscript (last paragraph of the introduction). In addition, as we mentioned above, the paper has been reviewed by the International Journal of Molecular Sciences English Editing Services.
1. Platelets cytoskeleton: tubulin
Line 55: Cytoskeleton contains spectrin, actin filaments and microtubules, actin and tubulin are building blocks.
Authors’ Response: We appreciated the reviewer comment. The sentence was not accurate. We have modified the manuscript in the following way: “Platelets cytoskeleton contains mainly spectrin, filaments formed by actin and microtubules formed by tubulins…” The revised statement can be found in page 2, beginning of the “Platelet cytoskeleton” chapter.
Line 56-57: What Authors understand by “shared or with more specific function”. This information is not clear.
Authors’ Response: We appreciated the reviewer comment. The sentence was finally removed.
Table 1.
Title: rather: “microtubule interacting proteins”
a-, b-, g- tubulins are not tubulin isoforms
dynein and dynactin are subunits of the complex
listed kinesins, MAPs, are not examples of isoforms.
Authors’ Response: We appreciated the reviewer comment. We have modified the manuscript in the following way:
A. Table 1 now is Table 2, titled “Major platelet microtubule interacting proteins.”
B. Isoforms column was removed.
C. Dynein and dynactin are considered as subunit of microtubule motor complex.
D. We removed the listed kinesins and MAPs from isoforms column.
Line 59: tubulin superfamily not family
Authors’ Response: We appreciated the reviewer comment. We have modified the manuscript and the following text has been added in Page 2, beginning of the “Platelets cytoskeleton” section, :“…tubulin are a widely expressed protein superfamily with GTPase activity”
Line 60: first, microtubules are implicated in the mentioned processes, not tubulin, second Authors mentioned “exocytosis”, but they probably meant “intracellular transport”.
Authors’ Response: We appreciated the reviewer comment. Accordingly, we remove all lines 60-61 from the original version (related to tubulin). Instead, all those functions have been associated to the microtubules. Thus, the following text has been added in “Platelets cytoskeleton” section, just before the current Table 1: “In eukaryotes, microtubules have an important role in processes such as structural support, intracellular transport, DNA segregation and cell motility”
Line 61-62: a,-b-g-tubulin are....in eukaryotic cells; ….tubulin can be found in other organisms, such as protists.
First, protists are Eukaryotes, second, for example d-tubulin is in centrosomes in mammals. So this sentence is incorrect and contradicts a next sentence (line 63).
Authors’ Response: We agree with the reviewer, and have clarified this in the manuscript. The revised statement can be found in the beginning of platelet cytoskeleton chapter, first paragraph.
Line 67: most abundant tubulin, not tubulin isoforms. Please see also below, and correct usage of “isoform” and “isotype”.
Authors’ Response: We fully agree with the reviewer. According to reviewer’s suggestion, we have modified “isoform” and “isotype”, as appropriate, throughout the review.
In particular, we have removed the previous statement “Among them, a, b, g, d and e-tubulin are the tubulin isoforms expressed in humans” by this one “These last ones (in reference to α, b and g-tubulin) are also the most abundant in human platelets, especially a and b tubulins”
Line 69-70: in case of tubulin the C-terminal end is called “tail” not “coil” (also line 75).
Authors’ Response: We appreciated the reviewer comment, and have modified “carboxy-terminal coil” to “carboxy-terminal tail”. The revised statement can be found in the 3º and 8º line from the fisrt paragraph of platelet cytoskeleton chapter.
Line 71: Copy number analysis of platelet proteins, by using …(remove “by”)
Authors’ Response: We appreciated the reviewer comment, and have modified the manuscript in the following way: “Copy number analysis of platelet proteins, by using proteomic studies,” was changed to “A copy number analysis of platelet proteins, using proteomic studies,” (Page 3, just after the first [ref 8]).
Line 75: cited REF 8 describes g-tubulin, not differences of the amino acid sequence of the tail of b-tubulin isotypes. Please cite proper REF. Also, line 70 (a-tubulin).
Authors’ Response: We thank the reviewer who has noticed there was a mistake with this reference. In the new version of the manuscript, we have cited the proper reference. The: current Ref 8 is: “Chakraborti, S.; Natarajan, K.; Curiel, J.; Janke, C.; Liu, J. The emerging role of the tubulin code: From the tubulin molecule to neuronal function and disease. Cytoskeleton 2016, 73, 521–550”.
Line 78: please see Strassel et al., 2019, Life Science Alliance (An essential role for α4A-tubulin in platelet biogenesis)
Authors’ Response: As pointed out by the reviewer, the previous statement “it remains unclear which α -tubulin isoform, if any, is the dominant form in platelets” was not right. We have followed the reviewer’s suggestion and modified the statment in this way: “It has been reported recently, however, that an α4A-tubulin enrichment in platelets and an increase in its expression occur during the late stages of MK differentiation” (Strassel et al., 2019; current ref 14).
Line 79: Microtubules are polymers built from dimers of a- and b-tubulin (protofibrils) that…
First, microtubule is composed of protofilaments not protofibrils. Second, it suggests that dimers are protofibrils. To form a tube there are two types of interactions between heterodimers (head-to-tail and side interactions).
Authors’ Response: We clarified in this revised version that “Dimers of α- and β-tubulin polymerize to form microtubules. Microtubules are, therefore, polymers built from heterodimers of α- and β-tubulin that connect in two ways (head-to-tail and side interactions) in order to form a rigid, hollow tube of approximately 25 nm in diameter.” The revised statement can be found in page 3, last paragraph, just before the current Table 1.
Line 81: 13 protofilaments are most common, but MTs with a different number of protofilaments also exist.
Authors’ Response: We agree with the reviewer, and have clarified this in the manuscript.
We have modified the manuscript in the following way: “Typically, the assembly of 13 protofilaments is required to form a single microtubule. However, the number of protofilaments per microtubule can vary”. The revised statement can be found in page 3, last paragraph, just before the current Table 1.
Line 83: dimer not dimmer
Authors’ Response: The suggested correction was taken into account and the text was modified accordingly.
Line 83-84: “The way that these subunits shape those polymers and their architecture, is different across species” – what Authors mean by this statement?
Authors’ Response: Since the paragraph where this statement was mentioned in the original version, was just related to the cytoskeleton in humans, this statement did not contribute anything. Therefore, we have eliminated it.
Line 84-86 – repetition. Importance of microtubules (as importance of “tubulin”) in diverse biological processes was mentioned earlier.
Authors’ Response: We appreciate this suggestion from the reviewer. Accordingly, we remove lines 60-61 from the original version, related to tubulin. In the current version, all those processes have been mentioned just once: Last line before the current Table 1.
Table 2: it is stated that it is a modification of work from REF 14 (Schwer et al., 2001). I can not find a Table in this REF.
Authors’ Response: We agree with the reviewer, and have changed the reference of that Table (in this version Table 1). The right reference is the current Ref 18. “Thomas, S.G. The Structure of Resting and Activated Platelets. In Platelets; Michelson, A.D., Ed.; Elsevier: Cambridge, MA, 2019; pp. 47–77 ISBN 9780128134566”.
Lines: 90-94:
Authors should be aware, that adding words” whereas” and “and” as well as changing where the sentence ends in somebodies text, is not proper scientific writing. In such a case, the text should be cited literally and taken in quotation marks. Otherwise, it could be considered plagiarism.
For comparison text from Burley et al., 2018
“Tubulinβ-1 chain is a 50.3 kDa protein containing 451 amino acids. The amino-terminal region (residues 1–206) mediates GTP binding and hydrolysis which is necessary for microtubule assembly. The intermediate region (residues 207–384) interacts with other tubulin molecules within the same or adjacent protofibrils. The carboxyl-terminal region (residues 385–45) mediates microtubule assembly and interactions with regulators [2,3].”
Authors’ Response: We appreciated the reviewer comment. In the current version of the review, when we cited literally, we have written the text between marks: “Tubulin b-1 chain is a 50.3 kDa protein containing 451 amino acids. Whereas, the amino-terminal region (residues 1–206) mediates GTP binding and hydrolysis (which is necessary for microtubule assembly), the intermediate region (residues 207–384) interacts with other tubulin molecules and the carboxyl-terminal region (residues 385–451) mediates microtubule assembly and interactions with regulators”.
Lines 95-97 – why these two proteins were mentioned (CAMSAP1 and MAP1)? Are they important in platelets? Publication cited by Authors refer to a general role of these proteins.
Authors’ Response: We appreciated the reviewer comment. We mentioned CAMSAP1 and MAP1 in a general context. MAP1 is also named in the current Table 2. To clarify the role of these proteins, we have added the following statement: “In platelets, the specific function of these proteins is unclear, but they seem to play a role in microtubule assembly and stability” (just before the current Table 2)
2. Role of tubulin in platelets formation
Line 109: “according to this hypothesis” instead of “In this theory”
Line 141: tubulin microtubules – change to microtubules (all microtubules are from tubulin)
Line 141: involving also…. Is “also” necessary?
Lines 143-146: The bidirectional transport may also be useful to mix platelet organelles, ensuring that each platelet obtains its portion, and maybe could explain the variability in the numbers of organelles that can be found within circulating platelets [31,32]. I am not sure about the usage of the word “mix” here.
Authors’ Response to the comments related to the lines 109, 141, 143-146: We appreciated the reviewer comment and all these points have been modified in the current version of the manuscript. Other spelling and grammar modifications have been done by the International Journal of Molecular Sciences English Editing Services. The revised statement can be found in the last paragraph of chapter 2.
3. Microtubules organization in resting platelets
Line 150:
Is: ….studies reveal that resting platelet microtubules are found around the circumference……form a marginal ring
Change to: ….studies reveal that in resting platelet microtubules are present beneath cell membrane around the circumference….where they form a marginal ring…
Authors’ Response: We appreciated the reviewer comment and have modified it in the manuscript. The revised statement can be found in page 4, first paragraph. In the current version and following the Reviewer 2’s suggestion we have moved the marginal ring explanation to the “Platelet cytoskeleton” chapter.
Line 153: microtubules “sit” (?), clear link – suggestion: an apparent link
Authors’ Response: We appreciated the reviewer comment and we have removed it ”Microtubules display occasionally intermicrotubule connections and sit below the spectrin membrane skeleton without making any clear link to the plasma membrane itself [33]”.
Line 155-156- repetition, please remove.
Authors’ Response: We have removes it in this version.
Lines 156 and next – refers to chapter 1.
Suggestion: please combine chapter 1 and chapter 3. Please clearly state that microtubules ring is also named microtubules coil (explain why coil – microtubules arrangement). In case of C-terminal end of tubulin use a term “tail”.
Authors’ Response: We have followed the reviewer’s suggestion and have combined chapter 1 with part of chapter 3. Similarly, we also incorporated the suggestion from the reviewer 2 to explain the “microtubule coils” concept in the first chapter of the manuscript (platelet cytoskeleton). The revised statement can be found in page 4, first paragraph.
5. Tubulin post-translational modifications in platelets
Line 189: remove: as we have just seen” What Authors mean by “MAPS can modulate this adaptation” – please be more precise.
General remark: please clearly separate general information regarding tubulin posttranslational modification and data concerning microtubules in platelets, cite appropriate publications. Publication about microtubules acetylation in resting cells is not cited here (REF 35, Patel-Hett et al., 2008).
Authors’ Response: In the “Tubulin post-translational modifications in platelets” chapter (previous chapter 5, but current chapter 4), we realized that the first paragraph was unclear. We have eliminated it. Instead, we have added, in the beginning of this chapter, general information about post-translational modifications. Then, in the next paragraph of this chapter we focus on tubulin posttranslational modification in platelets. We have added the Patel-Hett et al., 2008 reference ( ref 35) when we mentioned the microtubules acetylation.
Line 201-202: Enzymes involved in K40 deacetylation are HDAC6 (histone deacetylase family member 6).
Enzymes (two), HDAC6 and SIRT.
van Dijk et al., 2018, “Microtubule polyglutamylation and acetylation drive microtubule dynamics critical for platelet formation” It should be cited and role of glutamylation mentioned.
Authors’ Response: Following the reviewer’s suggestion we have added SIRT2 (sirtuin 2) in the new statement. We also mentioned the recent paper from van Dijk et al., 2018 and the role of role of glutamylation upon platelet activation. The revised statement can be found in the last lines of the second paragraph of this chapter.
6. Implications of tubulin genetic variability in pathological and physiological platelet function
Please explain what do you mean by rare variants, how it refers to tubulin mutation.
Authors’ Response: In the genetic field there is now an increasing tendency to substitute the term "mutation" for "rare genetic variant". It simply refers to a change in the gene sequence that is rare (MAF<1%). Indeed, the Standards and Guidelines for the Interpretation of Sequence Variants: A Joint Consensus Recommendation of the American College of Medical Genetics and Genomics and the Association for Molecular Pathology. Genet Med. 2015 May; 17(5): 405–424. doi:10.1038/gim.2015.30, there is recommendation for naming all changes in gen sequence as Variants and then sub classifying them as one of (1) pathogenic, (2) likely pathogenic, (3) uncertain significance, (4) likely benign, or (5) benign. However, in the current version, he have re-designated genetic variant as mutation or polymorphism depending on allele frequency (pag 7, chapter 5).
Lines 212-213:
…and loss of microtubule activity (absence of the platelet marginal band), but…
Loss of activity means that MTs are present but are not active (how Authors understand MTs activity, motor proteins activity on microtubules?). Absence of band / ring means that there are no MTs – if something is not present it can not be active. Please re-write.
Authors’ Response: We fully agree with the reviewer, and have re-written this statement in this way: “Although microtubules are decreased, disorganized and did not form platelet marginal bands, there is no abnormal bleeding” (current chapter 5 “Implications of tubulin genetic variability in pathological and physiological platelet function“, first paragraph).
Lines 233-238: Despite the absence of differences between neonatal and adult platelets under resting conditions, our group has recently shown that upon stimulation, MISSING TEXT ? while..
Sentence is very long. Please divide this sentence.
Authors’ Response: We have followed the reviewer’s suggestion and have divided it. This specific statement can be found, just before the Figure 3.
“Despite the absence of differences between neonatal and adult platelets under resting conditions, our group has recently shown that, upon stimulation, adult platelets exhibited dramatic alterations in shape and noticeable degranulation. Neonatal platelets, in contrast, displayed minor shape changes and their granules remained clearly visible throughout the cytoplasm, suggesting a limitation in the granule centering process”.
Lines 249-250: Since centralization of granules is a process that depends on contraction of the microtubule marginal coil,…..
Contraction or transport via motors, please explain (please cite REF)
Authors’ Response: Although the influence of the microtubules on granule release is relatively modest, after agonist exposure, cortical MT bundles contract leading to granule centralization before exocytosis. We appreciate the suggestion from the reviewer, and have incorporated two new references into the revised version of the manuscript (current REF 50 and 51).

Reviewer 2 Report
The manuscript provides a short overview over the role of microtubules in platelet biogenesis and in platelet activation, and it informs about a variety of mutations in the beta-tubulin 1 gene that are linked to pathological conditions in humans.
Although this manuscript may provide a useful review for non-specialists, I think it is poorly written and should be modified prior to acceptance.
In particular, I think that the topic is poorly introduced: the authors mention repeatedly the existence of “microtubule coils” in the first half of the manuscript, but without explaining them. The formation and significance of the “coils” is finally described in chapter 4 (line 164). I think that the readability of the manuscript would increase if the term “coil” was avoided in earlier paragraphs, or, alternatively, if the chapters on the formation and activation of platelets were moved to the beginning of the manuscript, followed by an in-depth description of tubulin isoforms and their mutations.
Other points:
- Abstract, line 16: alpha and beta-tubulin never exist as monomers in the cytoplasm; instead they form obligate heterodimers. The statement on monomers needs to be eliminated; in fact, it is in contradiction to the following text.
- Table 1: dynactin is not a motor. Dynactin is an accessory multiprotein complex that increases the processivity of the dynein motor.
- Line 103, correct “polyploid”, instead of “polyploidy”.
Author Response
The manuscript provides a short overview over the role of microtubules in platelet biogenesis and in platelet activation, and it informs about a variety of mutations in the beta-tubulin 1 gene that are linked to pathological conditions in humans.
Although this manuscript may provide a useful review for non-specialists, I think it is poorly written and should be modified prior to acceptance.
In particular, I think that the topic is poorly introduced: the authors mention repeatedly the existence of “microtubule coils” in the first half of the manuscript, but without explaining them. The formation and significance of the “coils” is finally described in chapter 4 (line 164). I think that the readability of the manuscript would increase if the term “coil” was avoided in earlier paragraphs, or, alternatively, if the chapters on the formation and activation of platelets were moved to the beginning of the manuscript, followed by an in-depth description of tubulin isoforms and their mutations.
Authors’ Response: We thank the reviewer his comment to improve the review organization. Thanks to this comment (coincident with the reviser 1) we realized that the review had to be reorganized in another way in order to make it understandable
To address comments from reviewers 1and 2; two major changes have been made.
A. We have moved the “microtubule coils” concept to the first chapter of the manuscript (platelet cytoskeleton); just after we provide information about the structure and function of the microtubules and following the Table 1. In this way, the “microtubule coils” concept is explained before it was mentioned.
B. We have reorganized the previous sections “Microtubules organization in resting platelets” and “Microtubules reorganization in platelets activation” into a new merged section “Microtubules organization in resting platelets and upon activation” (chapter 3 of the new version of the review):
Altogether, chapter 1 and Introduction have been modified deeply.
Other points:
- Abstract, line 16: alpha and beta-tubulin never exist as monomers in the cytoplasm; instead they form obligate heterodimers. The statement on monomers needs to be eliminated; in fact, it is in contradiction to the following text.
Authors’ Response: The reviewer´s comment is very right and, consequently, we have modified it in these terms:
“Two cytoskeletal polymer systems exist in megakaryocytes and platelets: actin filaments and microtubules, based on actin, and a- and b-tubulin heterodimers, respectively”.
- Table 1: dynactin is not a motor. Dynactin is an accessory multiprotein complex that increases the processivity of the dynein motor.
Authors’ Response: According to the reviewer's comment, we have modified the previous Table 1 (currently Table 2, page 4) it in these terms:
Dynactin, “A subunit of the microtubule motor complex that increases the processivity of the dynein motor”.
- Line 103, correct “polyploid”, instead of “polyploidy”
Authors’ Response: The suggested correction was taken into account and the text was modified accordingly. In addition, all manuscript has undergone English language editing by MDPI

Round 2
Reviewer 1 Report
The corrected version of the manuscript although significantly improved, still needs some additional work. Authors could shorten parts of the manuscript describing very basic textbook-like information about microtubules (e.g. lines 121-133) and focus more on data that describe microtubules and MIPs in platelets according to the aim of this review (e.g. lines 142 -152 provided information are very limited). Some information are repeated (probably due to text rearrangement) and such repetitions should be removed.
Finally, in a Summary Authors provided basically the same information as in an Abstract. One can learn again that MTs are composed of a-b- tubulin heterodimers, where are MTs in platelets, that these MTs are modified etc. - all we know from subtitles. What is missing is some conclusions, comments, putting this data in perspective.
Below I listed only some of the issues:
Lines 18-19
Is: Herein, we will focus on 1‐tubulin which, together with ‐tubulin, forms heterodimers which assemble into microtubules. Herein, we will
Suggested change: Herein, we will focus on platelet -specific tubulins and alterations and role of the MT skeleton in…/ during….
Lines 19-20
Is: During thrombopoiesis, microtubules mediate proplatelet elongation and
Suggested change: During thrombopoiesis, microtubules mediate elongation of the megakaryocyte extentions (propaltelet).. and
Line 23
Is: Molecular alterations in the encoding gene and post-translational …
Suggested change: Molecular alterations in the gene encoding b1 tubulin and microtubules post-translational…
Line 34-37
Is: …released into the bloodstream from the fragmentation of larger precursor cells called megakaryocytes (MKs). (and next sentence)
Suggested change: … released into the bloodstream as a result of fragmentation of larger precursor cells called megakaryocytes (MKs) that originate from hematopoietic stem cells. As MKs mature they become large and polyploidy.
Line 56
Suggestion –remove word “other”
Lines 55-57
Is; Platelets also play critical roles in other pathophysiological processes including , inflammation, vascular integrity, immunity, cancer metastasis and development [1].
Suggestion – perhaps move this sentence to the previous paragraph – I am not sure how to reorganize this part but I have an impression that lines 57-62 should be after line 53.
Lines 89-90
Is: “Platelet cytoskeleton contains mainly spectrin, filaments formed by actin and microtubules formed by tubulins.”
- Microtubules are always made of tubulin heterodimers.
Change:
Platelet cytoskeleton is mostly composed of spectrin, actin filaments and microtubules
Lines 91-92
“Focusing on the latter” – can be removed
Is: tubulin area widely…”
Change: tubulin are widely…
Line 95:
Suggested: ..can be found only in some…
Line 96:
Is: These last ones..
Suggested change: These last three tubulins…
Line 103:
Is: differences are higher…
Suggested: differences are more apparent, more prominent…
Lines 112-116:
Suggested change:
Mice / human b1 tubulin is composed of 451 amino acids and its theoretical molecular weight is 50.3 kDa. As in case of other beta tubulins, its N-terminal part encompassing 113 amino acids is involved in GTP binding and hydrolysis while middle part and C-terminal fragment are responsible for tubulin intermolecular interactions and binding of (some?) microtubule interacting proteins.
Comment: I am not sure if number of amino acids in each part of b1 tubulin is important for the reader. More interesting would be information of interacting proteins (called by Authors “ regulators) that were shown to act in platelets (if such are known).
Lines 121-133
These are very basic information and I believe can be omitted in such review. However, it would be interesting to know how many protofilaments are in MTs in platelets (13 as in case of other classical MTs or less or more ?). How dynamics of minus and plus MTs ends is regulated in platelets.
Lines 136-140
Is: Since platelets are anuclear “cells”, they lack a microtubule‐organizing center, which separates the chromosomes during cell division. In contrast, electron microscopy studies reveal that in resting platelets, microtubules are present beneath the cell membrane around the circumference of the discoid, in bundles of 3–24 microtubules, where they form a marginal ring, also named the microtubule coil [12,18], which maintains the discoid platelet shape and cell integrity in this state [11].
Suggested: Platelets are anuclear “cells” and they lack a microtubule‐organizing center. The electron microscopy studies reveal that in resting…
please re-write the remaining part of this sentence.
Table 1:
EB row Is: Binds to the microtubule plus its ends
Change: Binds to the microtubule plus ends
Is: Involved in a range of post-translational modifications including phosphorylation”. – please specify (how EB is involved in posttranslational modifications)
Line 178:
Is: microtubules move from the center to the cortex cell …
Was correct before : microtubules move from the center to the cell cortex … (cortical part of the cell not cells of the cortex)
Line 180:
Microtubule assembly occurs constantly inside the whole proplatelet, including
the shaft /axis, - I think shaft was correct
Line 185 – remove „thus”
Lines 201-203 have the same information as lines 199-201., please re-write entire part (lines 196-207)
Line 222: is: brunch of microtubules, please correct brunch
Line 281: is: tyrosination (in C‐terminal tyrosines). Modification by acetylation and tyrosination involves different mechanisms. During acetylation an acetyl group is added to lysine residue.
Tyrosination/ detyrosination – gene encoded a-tubulin in vast majority of cases has tyrosine in a last position. This tyrosine can be removed (detyrosination) and reattached (tyrosination). Thus, C-terminal tyrosine can’t be tyrosinated.
Authors do not mentioned so—called delta2 tubulin (removed two C-terminal amino acid and such modification was observed in platelets if I remember correctly).
Lines 287, 291 – remove “Thus”
Author Response
Reviewer #1 Comments and Suggestions for Authors
The corrected version of the manuscript although significantly improved, still needs some additional work. Authors could shorten parts of the manuscript describing very basic textbook-like information about microtubules (e.g. lines 121-133) and focus more on data that describe microtubules and MIPs in platelets according to the aim of this review (e.g. lines 142 -152 provided information are very limited). Some information are repeated (probably due to text rearrangement) and such repetitions should be removed.
Authors’ Response: We appreciate the comment on the shortening of paragraph included between lines 121-133, which only provides basic information of microtubules, and to provide more details in describing microtubules and MIPs in platelets. The suggested correction was taken into account and the text was modified accordingly.
· All changes in the new version are underlined in bold (black or red colour). Particularly the changes suggested by the reviewer on previous version in the lines 121-133 and 142 -152 can now be found in bold in the beginning of the second paragraph of Chapter 2 and in the last paragraph from the same Chapter.
Finally, in a Summary Authors provided basically the same information as in an Abstract. One can learn again that MTs are composed of a-b- tubulin heterodimers, where are MTs in platelets, that these MTs are modified etc. - all we know from subtitles. What is missing is some conclusions, comments, putting this data in perspective.
Authors’ Response: We appreciate this suggestion from the reviewer, and a new summary has been incorporated into the revised version of the manuscript.
Lines 18-19
Is: Herein, we will focus on b1‐tubulin which, together with a‐tubulin, forms heterodimers which assemble into microtubules. Herein, we will……
Suggested change: Herein, we will focus on platelet -specific tubulins and alterations and role of the MT skeleton in…/ during….
Authors’ Response: We appreciate this suggestion from the reviewer, and have incorporated it in the new version of the manuscript.
Lines 19-20
Is: During thrombopoiesis, microtubules mediate proplatelet elongation and
Suggested change: During thrombopoiesis, microtubules mediate elongation of the megakaryocyte extentions (propaltelet).. and
Authors’ Response: We appreciate this suggestion from the reviewer, and have incorporated it in the new version of the manuscript.
Line 23
Is: Molecular alterations in the encoding gene and post-translational …
Suggested change: Molecular alterations in the gene encoding b1 tubulin and microtubules post-translational…
Authors’ Response: We appreciate this suggestion from the reviewer, and have incorporated it in the new version of the manuscript.
Line 34-37
Is: …released into the bloodstream from the fragmentation of larger precursor cells called megakaryocytes (MKs). (and next sentence)
Suggested change: … released into the bloodstream as a result of fragmentation of larger precursor cells called megakaryocytes (MKs) that originate from hematopoietic stem cells. As MKs mature they become large and polyploidy.
Authors’ Response: We appreciate this suggestion from the reviewer, and have incorporated it in the new version of the manuscript.
Line 56
Suggestion –remove word “other”
Authors’ Response: We have removed it.
Lines 55-57
Is; Platelets also play critical roles in other pathophysiological processes including , inflammation, vascular integrity, immunity, cancer metastasis and development [1].
Suggestion – perhaps move this sentence to the previous paragraph – I am not sure how to reorganize this part but I have an impression that lines 57-62 should be after line 53.
Authors’ Response: The suggested correction was taken into account and the text was modified accordingly. Thus, the pathophysiological processes where platelets are important are moved to the previous paragraph (in the beginning to the second paragraph of the introduction, just before to explain that “their role in haemostasis and thrombosis”. Moreover, we have moved the previous lines 57-62 after line 53.
Lines 89-90
Is: “Platelet cytoskeleton contains mainly spectrin, filaments formed by actin and microtubules formed by tubulins.”
- Microtubules are always made of tubulin heterodimers.
Change:
Platelet cytoskeleton is mostly composed of spectrin, actin filaments and microtubules
Authors’ Response: The suggested correction was taken into account and the text was modified accordingly.
Lines 91-92
“Focusing on the latter” – can be removed
Is: tubulin area widely…”
Change: tubulin are widely…
Authors’ Response: The suggested correction was taken into account and the text was modified accordingly
Line 95:
Suggested: ..can be found only in some…
Line 96:
Is: These last ones..
Suggested change: These last three tubulins…
Authors’ Response: The suggested correction for lines 95 and 96 were taken into account and the text was modified accordingly
Line 103:
Is: differences are higher…
Suggested: differences are more apparent, more prominent…
Authors’ Response: We have changed “higher” for “more apparent”
Lines 112-116:
Suggested change:
Mice / human b1 tubulin is composed of 451 amino acids and its theoretical molecular weight is 50.3 kDa. As in case of other beta tubulins, its N-terminal part encompassing 113 amino acids is involved in GTP binding and hydrolysis while middle part and C-terminal fragment are responsible for tubulin intermolecular interactions and binding of (some?) microtubule interacting proteins.
Comment: I am not sure if number of amino acids in each part of b1 tubulin is important for the reader. More interesting would be information of interacting proteins (called by Authors “ regulators”) that were shown to act in platelets (if such are known).
Authors’ Response: We appreciate this suggestion from the reviewer, and modified the manuscript accordingly. In addition, the following text has been added (in the middle of first paragraph of Chapter 2):
· “Mice/human b1-tubulin is composed of 451 amino acids……….and binding of microtubule interacting proteins such us MAP2 (microtubule-associated protein 2) (Biochemistry. 1984 Sep 25;23(20):4675-81 & Biochemistry. 1991 Apr 30;30(17):4362-6)), which does steric inhibit dynein and kinesin motility (Cell Motil Cytoskeleton. 1993;24(1):1-16).While MAP2 is not expressed in human platelets, MAP1 is expressed and can also interact with C-terminal of b-tubulin (Biochemistry. 1991 Apr 30;30(17):4362-6).
Lines 121-133
These are very basic information and I believe can be omitted in such review. However, it would be interesting to know how many protofilaments are in MTs in platelets (13 as in case of other classical MTs or less or more ?). How dynamics of minus and plus MTs ends is regulated in platelets.
Authors’ Response: We understand the reviewer´s comment in order to remove basic information and highlight other aspects perhaps more interesting, such as the number of protofilaments or the dynamics of minus and plus microtubules ends is regulated in platelets. In this new version of the review we have added more information about the dynamics of the polymerization of the microtubules. The statement can be found in bold in the second paragraph of Chapter one.
Lines 136-140
Is: Since platelets are anuclear “cells”, they lack a microtubule‐organizing center, which separates the chromosomes during cell division. In contrast, electron microscopy studies reveal that in resting platelets, microtubules are present beneath the cell membrane around the circumference of the discoid, in bundles of 3–24 microtubules, where they form a marginal ring, also named the microtubule coil [12,18], which maintains the discoid platelet shape and cell integrity in this state [11].
Suggested: Platelets are anuclear “cells” and they lack a microtubule‐organizing center. The electron microscopy studies reveal that in resting… please re-write the remaining part of this sentence.
Authors’ Response: Following the suggestion of Reviewer, we have modified the paragraph accordingly. This new statement can be found just before Table2.
Table 1 (currently Table 2):
EB row Is: Binds to the microtubule plus its ends
Change: Binds to the microtubule plus ends
Authors’ Response: The suggested correction were taken into account and the text was modified accordingly.
Is: Involved in a range of post-translational modifications including phosphorylation”. – please specify (how EB is involved in posttranslational modifications)
Authors’ Response: We appreciated the reviewer comment in order to clarify the Table 2 content. The sentence was not accurate. Our intention was to mention that the regulatory effect of EB proteins on the microtubules polymerization depended on post-translational modifications in the EB protein itself, such us phosphorylation, acetylation, and detyrosination. To address the reviewer´s comment, we have modified the manuscript in the following way: “Binds to the microtubule plus ends to regulate microtubule activity, through post-translational modifications in EB protein itself, including phosphorylation, acetylation, and detyrosination (Cell Mol Life Sci. 2017 Jul;74(13):2381-2393).”
Line 178:
Is: microtubules move from the center to the cortex cell …
Was correct before : microtubules move from the center to the cell cortex … (cortical part of the cell not cells of the cortex)
Authors’ Response: We agree the reviewer comment. This change was suggested by the International Journal of Molecular Sciences English Editing Services. The statement has been modified to the original version.
Line 180:
Microtubule assembly occurs constantly inside the whole proplatelet, including the shaft /axis, - I think shaft was correct
Authors’ Response: We agree the reviewer comment. The statement has been modified to the original version.
Line 185 – remove „thus”
Authors’ Response: The suggested correction was taken into account and the text was modified accordingly
Lines 201-203 have the same information as lines 199-201., please re-write entire part (lines 196-207).
Authors’ Response: We fully agree with the reviewer, and have written entire part (lines 196-207). The revised statement can be found just before Chapter 3.
Line 222: is: brunch of microtubules, please correct brunch
Authors’ Response: The suggested correction was taken into account and the text was modified accordingly.
Line 281: is: tyrosination (in C‐terminal tyrosines). Modification by acetylation and tyrosination involves different mechanisms. During acetylation an acetyl group is added to lysine residue.
Tyrosination/ detyrosination – gene encoded a-tubulin in vast majority of cases has tyrosine in a last position. This tyrosine can be removed (detyrosination) and reattached (tyrosination). Thus, C-terminal tyrosine can’t be tyrosinated.
Authors do not mentioned so—called delta2 tubulin (removed two C-terminal amino acid and such modification was observed in platelets if I remember correctly).
Authors’ Response: We appreciated the reviewer´s comment. Effectively, the sentence for tyrosination was not accurate. On the other hand, delta2 tubulin has been described in platelets, as reviewer said.
To address the reviewer´s comment, we have modified the manuscript in the following way:
A. At the end of the previous line 281, we have added detyrosination/tyrosination processes (in C-terminal tyrosines).
B. We have included a new text after the previous line 281:
· “Additionally to last, another existing post-translational modification of α-tubulin, detected in platelet marginal band, is Δ2-tubulin (Cell 2018, 173, 1323–1327) which consists of removing sequentially the two C-terminal amino acids, i.e. tyrosine and glutamate (Cytoskeleton 2016, 73, 521–550). Thus, the previously tubulin detyrosination modification is required (J. Cell Biol. 2014, 206, 461–72) and once Δ2-tubulin happens, C-terminal cannot be retyrosinated (J. Cell Biol. 2014, 206, 461–72).”
Lines 287, 291 – remove “Thus”
Authors’ Response: The suggested correction was taken into account and the text was modified accordingly
